# What Mothers Know about Newborn Bloodspot Screening and the Sources They Use to Acquire This Knowledge: A Pilot Study in Flanders

**DOI:** 10.3390/children10091567

**Published:** 2023-09-18

**Authors:** Caroline di Gangi, Maren Hermans, Maissa Rayyan, François Eyskens, Karel Allegaert

**Affiliations:** 1Department of Public Health and Primary Care, Academic Centre for Nursing and Midwifery, 3000 Leuven, Belgium; digangicaroline@gmail.com (C.d.G.); maren.hermans@live.be (M.H.); 2Department of Development and Regeneration, KU Leuven, 3000 Leuven, Belgium; maissa.rayyan@uzleuven.be; 3Neonatal Intensive Care Unit, University Hospitals, UZ Leuven, 3000 Leuven, Belgium; 4Centre for Metabolic Diseases, University Hospital Antwerp, University of Antwerp, 2650 Edegem, Belgium; francois.eyskens@uza.be; 5Department of Pharmaceutical and Pharmacological Sciences, KU Leuven, 3000 Leuven, Belgium; 6Leuven Child & Youth Institute, KU Leuven, 3000 Leuven, Belgium; 7Department of Hospital Pharmacy, Erasmus University Medical Center, 3015 GD Rotterdam, The Netherlands

**Keywords:** newborn blood screening, knowledge, consent, mother, parent

## Abstract

To learn what mothers know about newborn bloodspot screening (NBS), the procedure, and the sources used, a pilot study was performed. An online questionnaire was developed, with the first part focused on characteristics and the NBS procedure, and the second on knowledge, information sources, and health care providers (HCPs). This questionnaire was accessible until 200 answers were received. The characteristics of respondents were representative for the population. Mothers gave verbal consent in 69.5% of cases, 12.5% did not, and 18% stated that no consent was requested. The ‘knowledge’ part contained 12 closed questions, five multiple-choice questions on sources, and assessments (5-point Likert scores) of the information transfer. The mean knowledge level was 7.2/12. Screening concepts (consequences, likelihood, sensitivity, carrier) and absence of notification of normal findings were well known. The fact that NBS is not compulsory was poorly known, and post-analysis sample handling procedures were poorly understood. Key HCPs were midwifes (80.5%) and nurses (38.5%). When the leaflet (44%) was provided, the majority read it. Mean Likert scores were 3.36, 3.38, 3.11 and 3.35 for clarity, timing appropriateness, sufficiency, and usefulness. The knowledge level and consent practices were reasonably good. Key HCP were midwives and nurses, the leaflets were supporting. This should enable a quality improvement program to a sustainable NBS program in Flanders.

## 1. Introduction

Newborn bloodspot screening (NBS) is one of the most-implemented population screening programs worldwide. While initially limited in terms of the number of diseases (phenylketonuria, hypothyroidism), NBS screening programs subsequently broadened their panel of conditions screened [1]. This was largely driven by technical improvements in diagnostics, and improved knowledge on disease mechanisms and history, converging with therapeutic interventions to improve the outcome in diagnosed infants. This progress relates to the fact that screening programs are driven by the criteria of Wilson and Jungner (an important health problem, the natural history of the condition is well understood, it is detectable at an early stage, earlier treatment should be beneficial, a suitable test should be available in this early stage (sensitivity/specificity), and the test should be acceptable) [1]. Although not causally related, this expansion of NBS programs also raised awareness of the shortcomings in parental education, information products, and the informed consent process. This can, in part, also relate to the fact that expansion of the NBS programs will also result in communications to the different stakeholders involved, and may induce reflections or questions. The provision of information to parents has been recognized as a crucial part of sustainable NBS programs, being that there is still lack of regulatory harmonization within Europe [2].

Almost all European countries provide information for parents (brochures, websites). About two-thirds of the countries ask for consent, while consent for long-term storage of blood spot cards is requested in a minority (30%) of European countries [1]. In a recent paper, Ijzebink et al. focused on these information products provided throughout Europe [3]. In this paper, 26 printed European products (like leaflets or flyers) were assessed on their content and knowledge, and rated according to a list of eight knowledge domains (screening purpose, false positive/negative findings, uncertainties and risks, medical implications, social implications, financial implications, follow-up, and support services). Despite some differences between European countries, most of these eight knowledge aspects were included in all information products, with most diversity related to the handling of residual bloodspot samples [3].

Along the same lines, the script for health care providers (HCPs) involved in NBS in Flanders (in the north of Belgium; the Dutch-speaking part) mentions the need to discuss the relevance of timely diagnosis, the importance of the postnatal age window (72–96 h) for screening, the practicalities related to the appointment for the screening (because of the short hospital stay after delivery), the need of verbal (concise) consent, and to make it clear that all initial screening costs are covered by the government, costs for additional tests after screening are reimbursed by the insurance, which diseases are screened for and, finally, the fact that NBS is not compulsory, but highly recommended [4]. In Flanders, long-term storage of the blood spot charts is not part of the program, and storage is limited to one year. These topics are also discussed in leaflets (printed, and as a downloadable pdf) and on a specific website (aangeboren.bevolkingsonderzoek.be) to inform both parents and the general public [5].

Informed consent necessitates that the relevant person(s) have been informed on the procedure and its potential consequences in such a way that it is reasonable to assume that the information has been understood sufficiently well to support their decision (irrespective of the subsequent decision itself) [6]. In the study by Frankova et al., 12/27 countries mentioned that checks (commonly based on surveys) are performed to verify that information indeed reaches the targeted populations [2]. In an attempt to obtain a first snapshot on what mothers know on NBS, and the sources they use in Flanders to acquire this knowledge (as the Flanders region or Belgium as a country were not included in European survey) [2], a pilot study was performed.

## 2. Materials and Methods

### 2.1. Perinatal Health Care Structures in Flanders and the Study Setting

Prenatal follow up in Flanders is mainly coordinated by obstetricians and midwives, with more limited involvement of general practitioners, and is still almost exclusively ‘hospital’-driven. In 2021, there were 63,334 (64,282 births) deliveries in Flanders. Those deliveries almost exclusively occurred in one of the 59 hospitals (‘maternities’), with about 0.8% of deliveries elsewhere (like at home, or in birth centers not connected to a hospital). In an attempt to evolve to a transmural care program, initiatives were taken to shorten the duration of hospitalization to 2–4 days, in part depending on the type of delivery. This shift has been facilitated by the development of midwife-driven home care programs, supported by general practitioners, obstetricians, and pediatricians [7,8]. Consequently, a relevant portion of newborns have their NBS screening performed at home by midwives, and different HCPs are commonly involved in the pregnancy and postpartum care.

Within this health care framework, and as a pilot study, we aimed to recruit 200 mothers who recently (maximum 1 year before the dissemination of the questionnaire) delivered in Flanders, understood Dutch sufficiently well to complete the questionnaire, were older than 18 years, and provided consent to contribute to the pilot study. We are aware that with this approach, partners of the mothers were a priori excluded. We are aware that this is a deficiency, but we deliberately had to do this to avoid ‘dual’ reporting within the framework and limitations of this pilot study.

### 2.2. Questionnaire

In the first part of the questionnaire, information on characteristics and background of respondents (like age, residency, primi- or multipara, level of education, place of delivery) and on the NBS procedure (collected yes/no), location (hospital or at home) of the procedure, consent recall) were collected. The second part focused on the knowledge itself, the sources of information used, and the HCPs involved. This part of the questionnaire was constructed in line with the approach described by Detmar et al. for a Dutch cohort, with some adaptations to the Flemish setting (health care organization, legal and regulatory environment, sources of information) [6]. To respect the methodology on questionnaire design, adaptations were initially made independently by CdG and MH, with subsequent cross-verification to attain consensus. In the event of absence of consensus, KA was involved. The final version was subsequently verified on face validity by these three authors.

### 2.3. Data Collection and Analysis

Questionnaires were distributed online (Qualtrics, Seattle, WA, USA), using social media platforms (Facebook, personal and group pages) and e-mail correspondence to nurseries. To assess the representativity of respondents, information on characteristics and background was compared to reference information on pregnancies in Flanders [7,9]. Data on the NBS procedure (collected yes/no, location of the procedure, consent recall) were more difficult to compare, as we could only retrieve a press release from the relevant agency that stated that 99% of the newborns undergo NBS in Flanders [10]. Data analysis on knowledge was based on maternal knowledge on the NBS procedure as the dependent variable; the independent variables were the information received (as perceived by the mother), parity (prima- versus multigravida), and the level of maternal education. We hereby a priori hypothesized that maternal knowledge would correlate positively with parity and the level of education.

### 2.4. Ethics, Privacy, and Data Management 

The Ethics Committee Research of KU Leuven and University Hospitals Leuven approved the study protocol (MP022668, 5 December 2022, favorable advice). The questionnaire was preceded by an information letter, describing the aims of the study and the consent to contribute to the questionnaire. Consent to contribute and store responses for analysis (anonymous, confidential) was requested before the questionnaire could be completed.

## 3. Results

The final version of the ‘knowledge’ part (translated version, English, Table 1) contained 12 closed questions, and five multiple choice + open questions on the sources of information. We have provided the Dutch version of the questionnaire in a Appendix A to facilitate future use.

Finally, based on a 5-point Likert score, respondents were requested to provide their assessment of the information transfer (clarity, appropriateness of timing, sufficiency, usefulness) process.

This questionnaire (characteristics and background, NBS procedure and knowledge part, and the general assessment of knowledge transfer process) was accessible online from 2 February 2023 to 18 April 2023, when 200 questionnaires were received.

Participants were recruited by their own Facebook profiles (CdG, MH, n = 42), specific groups within Facebook (six groups, 709 messages, 100 participants), or nurseries (n = 79 nurseries contacted). Ninety-eight % of the participants finalized the questionnaire. To assess representativity of respondents to the Flemish pregnant and postpartum population, age, residence (postal code, provinces), parity, level of education, place of delivery (hospital, or out of hospital), collection of the NBS (yes/no; location (hospital/home)) of respondents were compared to the latest Perinatal Epidemiology Study Center (SPE) 2021 annual report and STATBEL report (education) (Table 2). Maternal age and place of delivery were similar, while there were some differences in place of residence, some overrepresentation in primigravidae, and respondents had a somewhat higher level of education compared to the reference population. Data on the NBS procedure (yes/no, place, verbal consent) were somewhat more difficult to compare with the reference population data, but also seem similar (Table 2). NBS were collected both in the hospital, as well as at home. Mothers recalled verbal consent in 69.5% of cases, 12.5% did not recall any consent request, and 18% stated that no consent has been requested.

Based on the 12 questions provided, the mean level of knowledge was 7.2 (SD 2.4)/12, and 79% of the respondents had a score ≥6. The level of knowledge was correlated positively with the level of education, and without a difference between primi- and multipara. An overview on the responses to the individual questions is provided in Table 3.

The concepts of targeted screening (severe consequences, low a priori likelihood, sensitivity, carrier concept) and absence of notification in the event of normal findings are well known. In contrast, the fact that NBS is not compulsory in Flanders is only poorly known, and the post-analysis handling of the NBS sample is poorly understood.

Related to the sources of information, the most relevant HCPs involved were midwifes (80.5%) and nurses (38.5%), while other sources were obstetricians (20%), the leaflet (12%), or general practitioners (1.5%). Five percent did not recall having received any information on the NBS procedure, 5.5% of the respondents mentioned that they had already received information on NBS during their nursing or medical training. A similar pattern was observed on the question to indicate the most relevant source(s) of information, with verbal interaction with HCPs (midwifes, 77.9%; nurses 30.7%; obstetricians 18.1%) superior to the information leaflet (7.5%). Forty-for percent of the mothers reported that they received the leaflet. Of those who received the NBS leaflet, the majority had read the leaflet, either completely (34%) or at least partially (87.5%).

Finally, and based on a 5-point Likert score, respondents provided their general assessment on the information transfer on *clarity* (3.36, SD 1.22), *appropriateness of timing* (3.38, SD 1.46), *sufficiency* (3.11, SD 1.6), and its *usefulness* (3.35, SD 1.29). A significant positive correlation was observed between the individual respondent’s knowledge score and the Likert score.

## 4. Discussion

We report on what mothers know on NBS, and their sources of information in Flanders. This provides the first snapshot of the overall knowledge (mean level 7.2/12 questions) and the most relevant sources of persons involved (midwives, nurses). When the leaflet (44%) was provided, the majority had read it at least partially. The concepts of targeted screening and absence of notification in the event of normal findings were well known. In contrast, the fact that NBS is not compulsory in Flanders was only poorly known, and the post analysis handling of the NBS sample was poorly understood (Table 3). Finally, the overall Likert rating on knowledge transfer was reasonably (3.3/5) good.

Related to the representativity and feasibility, we wanted to stress that, despite some minor differences (Table 1), this pilot cohort largely represents the overall population of mothers who recently gave birth in Flanders. Furthermore, the majority of the respondents finalized the questionnaire, suggesting that the burden (time, type of questions) was perceived to be reasonable and relevant. Furthermore, the location of NBS (hospital/home) sampling likely also reflects contemporary practices.

On maternal knowledge, there was good to very good performance on the concepts of targeted screening (severe consequences, low a priori likelihood, sensitivity, carrier concept), and absence of any notification in the event of normal findings. In contrast, the fact that NBS is not compulsory in Flanders is only poorly known, and the post analysis handling of the NBS samples (destroyed after one year, but not ‘out of scope’ clinical research allowed) is poorly understood (Table 3). Procedurally, mothers recalled verbal consent in 69.5% of cases, 12.5% did not recall any consent request, and 18% stated that no consent has been requested. The results on both knowledge and consent practices are similar to somewhat better, compared to other recently reported surveys on this topic [11,12,13]. Key HCPs for this knowledge transfer are midwives and nurses, with the leaflet as a supporting resource [11,14].

In terms of question 3 (“for the NBS, some blood is collected from the heel of the infant”), we wanted to explore the specific knowledge of parents on the ‘irrelevance’ of the site of blood collection in itself, as ‘heel lancing = hielprik’ is perceived and used as a synonym for NBS, which refers to the blood sampling rather than the anatomic site. In terms of question 12 (“are you aware of the recent extension of diseases screened for with the NBS?”), this referred to very recently implemented screening for spinal muscular atrophy (SMA) in Flanders, as this implementation was associated with specific campaigns focused on both HCPs and the general public. Obviously, this study has relevant limitations. Besides the pilot character and exclusive focus on mothers, this study design obviously holds the risk of recall bias (in both directions). One could also reflect on the completeness of the questionnaire, as, post hoc, not all eight previously mentioned highlighted knowledge domains (e.g., financial implications) were sufficiently well-covered [3,15]. Still, we feel that there is value in this pilot study beyond feasibility, as it is relevant to regularly check that information indeed reaches the targeted populations, and that practices remain concordant (like relevant portion of mothers that do not recall a verbal consent request) [2]. We therefore suggest a quality improvement cycle towards a sustainable NBS program, with regular updated surveys as part of this strategy (Figure 1).

Such a program should be driven by well-trained HCPs (knowledge, communication skills, and practices) so that the correct, relevant information can be provided, with access to the updated information of NBS practices, as described in the above-mentioned HCP script on NBS [4]. HCPs’ knowledge and skills training should focus on the relevant information to be provided, to avoid overload [16]. Based on other research, this is preferably performed during pregnancy, to be verified in early postpartum, with midwives or nurses in the lead [6,17]. These HCPs are also reported by the mothers as the key persons involved in knowledge transfer, in line with similar reports [18]. The recall of verbal consent in only 69.5% of the mothers suggests that any quality improvement program should also reinforce the verbal consent practice as part of the NBS procedure. The impact, strengths, and potential weaknesses can subsequently be assessed by regular surveys, as done in this pilot. However, we do believe that there is value in co-creating the next version of such a questionnaire in collaboration with HCPs, the agency involved, and the public. Any optimization will likely be along the eight domains recently identified in a European survey [3,15]. Similarly, a recent French qualitative study on parental information and consent also listed five themes (knowledge, information received, parental choice, experience of the NBS process, and parents’ perspectives and wishes) [14].

Such a co-creation model is likely also beneficial for the knowledge sources (website, leaflet). Finally, and although this was not part of the current study, we do believe that informing the general public by websites or media is an effective additional approach to ensure sustainable NBS practices. At least, we hope this pilot study, considering all the limitations of a pilot study, has paved the way to implement such a quality improvement program to attain a sustainable NBS program in Flanders.

## Figures and Tables

**Figure 1 children-10-01567-f001:**
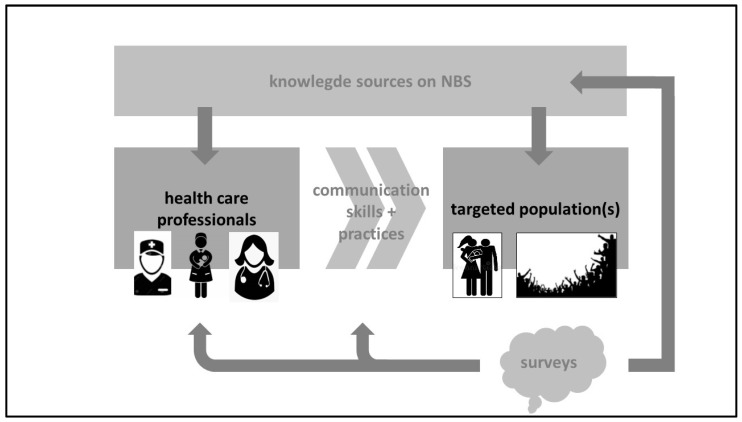
Schematic overview of the suggested quality improvement cycle, illustrating how surveys can have impact on knowledge sources (for both health care providers and the public), as well as on the training of these health care providers (communication skills and practices).

**Table 1 children-10-01567-t001:** The final questionnaire.

**Knowledge questions, closed**
Diseases screened for with the NBS have severe consequences if not treated appropriately.
The likelihood that an infant has a disease screened for is low.
For the NBS, some blood is collected from the heel of the infant.
In the event of an abnormal NBS, additional investigations in the hospital are needed.
A normal NBS provides certainty that the infant is perfectly healthy.
The NBS test is reliable, as an infant with a given disease screened for will very likely be detected.
In the event of uncertainties, a second NBS is indicated.
The NBS is compulsory for any newborn.
A healthy person can still be the carrier of a genetic disease.
When the NBS is NORMAL, parents will NOT receive a notification.
Immediately after the NBS analysis, the blood sample will be destroyed.
Are you aware of the recent extension of diseases screened for with the NBS?
**Knowledge questions, multiple choice, including open answers**
Who was involved to inform you about the NBS (you can provide multiple answers)? (*midwife, nurse, obstetrician, general practitioner, information session during pregnancy, friends or peers, website ‘aangeboren.bevolkingsonderzoek.be’, television or radio, journals or magazines, the NBS folder, social media, I have not received information, others*)
What was the most relevant source of information on NBS for you (you can provide multiple answers)? (*midwife, nurse, obstetrician, general practitioner, information session during pregnancy, friends or peers, website ‘aangeboren.bevolkingsonderzoek.be’, television or radio, journals or magazines, the NBS folder, social media, I have not received information, others*)
If you have received the NBS folder, have you read this document? (*yes, I have read this document fully, partial, screened; no, as I was already aware of the folder, or I already knew on the NBS, not applicable as no folder received*).
Have you searched for other sources of information? (*no; yes, on the website ‘aangeboren.bevolkingsonderzoek.be’; yes, on the internet, but other websites; yes, other folders, books or magazines; yes, I have discussed this with others; yes, as…*)
How do you overall assess the information you have received? (Likert score 0–5, where 0 is the worse score) (*clarity, appropriateness of the timing, sufficiency, usefulness*)

NBS: newborn bloodspot screening.

**Table 2 children-10-01567-t002:** Representativity of the respondents to the Flemish pregnant and postpartum population (NBS: newborn blood screening) [8,9].

Variables	Categories	Respondents	Reference Population
**Maternal age**	<20 years	0.5%	0.9%
	20–24	13.5%	8.1%
	25–29	39%	32%
	30–34	34%	40.1%
	35–39	9%	15.4%
	≥40	4%	3.5%
	Brussels	0.5%	unknown
**Residence**	Brabant, Flemish	15%	14.4%
	Antwerp	28.5%	34.2%
	Limburg	34%	10.9%
	West Flanders	9%	17.4%
	East Flanders	13%	23.1%
**Parity**	Primipara	59%	45.2%
	Multipara	41%	54.8%
**Education**	Bachelor onwards	61.5%	56.2%
	≤High school	38.5%	43.8%
**Place of delivery**	Hospital	97%	96%
	Out of hospital	3%	4%
**NBS collected**	Yes	99%	>99%
	No	0.5%	
	Unclear	0.5%	
**Place of NBS**	Hospital	54%	unknown
	Home	46%	unknown

**Table 3 children-10-01567-t003:** Overview of the answers (%) received for the 12 questions on neonatal blood screening (NBS) knowledge. The correct answers are highlighted in grey.

Questions	Yes	No	Do Not Know
Diseases screened for with the NBS have severe consequences if not treated appropriately.	71.5%	3%	25.5%
The likelihood that an infant has a disease screened for is low.	51%	11.5%	25.5%
For the NBS, some blood is collected from the heel of the infant.	26%	72%	1.5%
In the event of an abnormal NBS, additional investigations in the hospital are needed.	73%	3%	22.5%
A normal NBS provides certainty that the infant is perfectly healthy.	5%	86.5%	8.5%
The NBS test is reliable, as an infant with a given disease screened for will very likely be detected.	57.5%	6%	36.5%
In the event of uncertainties, a second NBS is indicated.	44.5%	9%	46.5%
The NBS is compulsory for any newborn.	38%	40%	21.5%
A healthy person can still be the carrier of a genetic disease.	88.5%	0.5%	11%
When the NBS is NORMAL, parents will NOT receive a notification.	89%	6%	4.5%
Immediately after the NBS analysis, the blood sample will be destroyed.	12%	9%	79%
Are you aware of the recent extension of diseases screened for with the NBS?	61.5%	38.5%	0%

## Data Availability

The corresponding author can be contacted if additional information is required.

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
