# Peer review of "What Mothers Know about Newborn Bloodspot Screening and the Sources They Use to Acquire This Knowledge: A Pilot Study in Flanders"

_children, 2023, doi:10.3390/children10091567_

Round 1
Reviewer 1 Report
The authors present an overview of mothers' knowledge on newborn screening in Flanders. The article is pleasant to read, well presented, the research is well conducted and the subject, while not new, is interesting to explore. I would like to congratulate the authors on their excellent work.
For information purposes, it might be interesting for the reader to know, and I would have liked to know:
1. Where is the blood is taken in Flanders (in the questionnaire, the question is asked about heel sampling, which is still very much in the collective imagination, including that of doctors)?
2. What is the extension of the screening programme mentioned in the questionnaire? And more generally, what diseases are screened for?
Be careful not to transcribe in the discussion all the results presented a few lines above in the results section, as the discussion of these results does not need to be repeated in full.
A few minor comments on the introduction, which sometimes makes inappropriate links:
- L42/43: "This progress reflects the fact that..." there is progress, which follows these recommendations, but there is no causal link
- L46-47: "This expansion of NBS programs also raised awareness » just because the number of diseases screened is increasing does not mean that parents are better informed
- L53-54: sentence incomplete: "A recent paper focused on these information products provided".
- L55: "printed products" to be replaced by Flyer or liflet
- L55: "compared". The term is unsuitable, perhaps "rated according to...".
- Perhaps specify "Flanders, in northern Belgium."
Author Response
The authors present an overview of mothers' knowledge on newborn screening in Flanders. The article is pleasant to read, well presented, the research is well conducted and the subject, while not new, is interesting to explore. I would like to congratulate the authors on their excellent work.
WE THANK THE REVIEWER FOR THE VERY SUPPORTIVE AND POSITIVE ASSESSMENT OF OUR WORK. WE ACKNOWLEDGE, AS ALSO ALREADY MENTIONED IN THE PAPER THAT THIS WORK IS NOT NEW AT THE INTERNATIONAL LEVEL, BUT IS INDEED NEW INFORMATION FOR FLANDERS AND THEREFORE ADD TO THE EXISTING LITERATURE.
For information purposes, it might be interesting for the reader to know, and I would have liked to know:
- Where is the blood is taken in Flanders (in the questionnaire, the question is asked about heel sampling, which is still very much in the collective imagination, including that of doctors)?
THERE IS NO STRONG GUIDANCE ON THE LOCATION OF SAMPLING SO THAT WE INTEND TO EXPLORE THESE ‘BEDSIDE’ PRACTICES INDEED IN THE NEAR FUTURE TO GET A BETTER IDEA. BASED ON THE BACKGROUND OF THE CORRESPONDING AUTHOR IN THE PAIN OF NEONATAL PAIN PREVENTION, WE WOULD HOPE THAT THERE VENOUS PUNCTURE ARE MUCH MORE COMMONLY APPLIED, IN COMBINATION WITH NON-PHARMACOLOGICAL STRATEGIES TO COOP WITH THE PROCEDURAL PAIN. AT PRESENT, WE DO NOT REALLY KNOW, BUT THE HEEL LANCING IS PERCEIVED BY PARENTS AS A SYNONYM TO GUTHRIE OR NBS SAMPLING.
- What is the extension of the screening programme mentioned in the questionnaire? And more generally, what diseases are screened for?
WE HAVE ADDED THIS INFO, BUT IN ESSENCE, THERE IS AN ONGOING PLAN TO EXTEND THE NBS SCREENING, WITH MOST RECENT SMA AND IMPLEMENTED SINCE A FEW YEARS, CF SCREENING. BOTH NOVELTIES HAVE RESULTED IN PUBLIC CAMPAINGS, AND SUCH CAMPAINGS MAY INDUCE REFLECTIONS BY BOTH CARE PROVIDERS AND THE PUBLIC ON THE SCREENING PRACTICE.
Be careful not to transcribe in the discussion all the results presented a few lines above in the results section, as the discussion of these results does not need to be repeated in full.
WE HAVE CONSIDERED YOUR SUGGESTION ON THE DISCUSSION SECTION AND HAVE ADAPTED.
A few minor comments on the introduction, which sometimes makes inappropriate links:
- L42/43: "This progress reflects the fact that..." there is progress, which follows these recommendations, but there is no causal link
- L46-47: "This expansion of NBS programs also raised awareness » just because the number of diseases screened is increasing does not mean that parents are better informed
- L53-54: sentence incomplete: "A recent paper focused on these information products provided".
- L55: "printed products" to be replaced by Flyer or liflet
- L55: "compared". The term is unsuitable, perhaps "rated according to...".
- Perhaps specify "Flanders, in northern Belgium."
WE HAVE CONSIDERED YOUR SUGGESTION ON THE INTRODUCTION SECTION AND HAVE ADAPTED. WE CONFIRM WE NEVER INTENDED TO SUGGEST CAUSALITY (L46-47), AND HAVE ADDED SOME REFLECTIONS ON THIS IN THE RELEVANT SECTION OF THE PAPER.

Reviewer 2 Report
I found the paper to be of general interest and generally well presented. Aside from some English issues (below), I am confused by the third line in Table 3. Since blood is collected from a heel stick, it would seem that the gray area should be in the 'yes' column and not the 'no' column as shown. Are these responses transposed?
There are some typos and awkward wording, for example line 3 in Table 1 - 'form' when it should be 'from,' and again in Table 3. Also in Table 3 - line 1, use 'appropriately' instead of 'appropriate' ; eliminate 'anyhow' in line 2. Line 169 should be - "Five percent did not recall having received ..." Line 188, "...the majority read it, at least partially." There are other examples throughout the paper that need some correcting, mainly tenses.
Author Response
I found the paper to be of general interest and generally well presented. Aside from some English issues (below), I am confused by the third line in Table 3. Since blood is collected from a heel stick, it would seem that the gray area should be in the 'yes' column and not the 'no' column as shown. Are these responses transposed?
There are some typos and awkward wording, for example line 3 in Table 1 - 'form' when it should be 'from,' and again in Table 3. Also in Table 3 - line 1, use 'appropriately' instead of 'appropriate' ; eliminate 'anyhow' in line 2. Line 169 should be - "Five percent did not recall having received ..." Line 188, "...the majority read it, at least partially." There are other examples throughout the paper that need some correcting, mainly tenses.
WE ALSO WOULD LIKE TO THANK THE SECOND REVIEWER FOR THE VERY SUPPORTIVE AND POSITIVE ASSESSMENT OF OUR WORK. WE HAVE CONSIDERED YOUR SPECIFIC TEXTUAL ASUGGESTIONS AND HAVE ADAPTED THIS IN THE REVISED VERSION.
RELATED TO THE QUESTION 3, THIS HAS ALSO BEEN RAISED BY THE FIRST REVIEWER AND WE HAVE ELABORATED ON THIS IN THE DISCUSSION PART OF THE PAPER.

Reviewer 3 Report
The study by de Gangi et al. addresses an important issue related to newborn screening, i.e. information of parents (mothers in this case) about NBS. The paper is well-written and data suggest that in particular the verbal communication with HCP, mainly midwives in Flanders, and thus the information and training of these HCP are crucial for transfer of information. This is a perception worth to be distributed to the (noteworthy small) NBS community.
However, though the limitations are well discussed, it cannot be neglected that the limitations of this paper are severe. Firstly, the entire paper is based on a very small and simple questionnaire Furthermore, extrapolation from a relatively small and unique region like Flanders to other (European) regions or countries are difficult. Second, taken in account that there are 59 hospital-based birth wards and numerous midwives involved in home care programs, a number of 200 can hardly be representative, even though data on representativity is given. Of note, authors correctly address this shortcoming by repeatedly mentioning the study to be a pilot study.
Author Response
The study by de Gangi et al. addresses an important issue related to newborn screening, i.e. information of parents (mothers in this case) about NBS. The paper is well-written and data suggest that in particular the verbal communication with HCP, mainly midwives in Flanders, and thus the information and training of these HCP are crucial for transfer of information. This is a perception worth to be distributed to the (noteworthy small) NBS community.
WE ALSO WOULD LIKE TO THANK THE THIRD REVIEWER FOR THE VERY SUPPORTIVE AND POSITIVE ASSESSMENT OF OUR WORK.
However, though the limitations are well discussed, it cannot be neglected that the limitations of this paper are severe. Firstly, the entire paper is based on a very small and simple questionnaire Furthermore, extrapolation from a relatively small and unique region like Flanders to other (European) regions or countries are difficult. Second, taken in account that there are 59 hospital-based birth wards and numerous midwives involved in home care programs, a number of 200 can hardly be representative, even though data on representativity is given. Of note, authors correctly address this shortcoming by repeatedly mentioning the study to be a pilot study.
WE AGREE THAT – BEING A PILOT STUDY – THAT THE EXTRAPOLATION AND INTERPRETATION HAS MAJOR LIMITATIONS, BUT AS LEAST WE WANTED TO SUPPORT THE REPRESENTATION OF THE PILOT RESPONDENTS TO THE FLEMISH SETTING. AS ALREADY MENTIONED IN THE INITIAL VERSION, AND FURTHER EXTENDED IN THE REVISED VERSION, WE HOPE THAT THIS PILOT WILL BE FOLLOWED BY A MORE EXTENSIVE SURVEY THROUGHOUT FLANDERS. HOWEVER, TO ENABLE THIS, ADDITIONAL FUNDING WILL BE NEEDED BESIDES A REACH OUT TO THE DIFFERENT STAKEHOLDERS. THIS WAS ALREADY MENTIONED IN THE INITIAL VERSION OF THE PAPER, AND HAS BEEN SOMEWHAT FURTHER STRESSED IN THE REVISED VERSION IN THE CONCLUSION SECTION.

Round 2
Reviewer 3 Report
No further comments.